# Critical Roles of METTL3 in Translation Regulation of Cancer

**DOI:** 10.3390/biom13020243

**Published:** 2023-01-27

**Authors:** Wangyang Meng, Han Xiao, Peiyuan Mei, Jiaping Chen, Yangwei Wang, Rong Zhao, Yongde Liao

**Affiliations:** 1Department of Thoracic Surgery, Union Hospital, Tongji Medical College, Huazhong University of Science and Technology, Wuhan 430022, China; 2Department of Thoracic Surgery, Ruijin Hospital, Shanghai Jiao Tong University School of Medicine, Shanghai 200000, China

**Keywords:** METTL3, m^6^A, translation regulation, post-transcriptional modification, eukaryotic initiation factor, RNA binding protein, cancer

## Abstract

Aberrant translation, a characteristic feature of cancer, is regulated by the complex and sophisticated RNA binding proteins (RBPs) in the canonical translation machinery. N6-methyladenosine (m^6^A) modifications are the most abundant internal modifications in mRNAs mediated by methyltransferase-like 3 (METTL3). METTL3 is commonly aberrantly expressed in different tumors and affects the mRNA translation of many oncogenes or dysregulated tumor suppressor genes in a variety of ways. In this review, we discuss the critical roles of METTL3 in translation regulation and how METTL3 and m^6^A reader proteins in collaboration with RBPs within the canonical translation machinery promote aberrant translation in tumorigenesis, providing an overview of recent efforts aiming to ‘translate’ these results to the clinic.

## 1. Introduction

N6-methyladenosine (m^6^A) modifications in RNA were first identified in the 1870s [1]. The enzyme that catalyzes the formation of m^6^A is known as m^6^A “writer”, methyltransferase-like 3 (METTL3), which is the only catalytic subunit of the methyltransferase complex and can synthesize almost all m^6^A modifications in mRNAs [2]. As research progressed, it was gradually realized that m^6^A modifications are an essential regulatory modality in biological development, affecting cell differentiation and other physiological processes. Since the rise of m^6^A high-throughput sequencing methods in 2012 [3,4], more and more studies have found that the epitranscriptome plays a key role in regulating the fate and function of mRNAs in cells.

m^6^A is a selective modification enriched in specific mRNAs [5]. Some mRNAs contain only a single m^6^A site, but some contain 20 or even more m^6^A sites [4]. Overall, about 50–80% of mammalian mRNAs may m^6^A sites be absent [4,6,7,8]. Under physiological conditions, m^6^A is enriched in the 3’ untranslated region (3’UTR) and near the stop codons of the transcripts [3]. Analysis of mRNAs enriched in m^6^A modifications showed enrichment of developmental regulation and cell fate-related genes [9]. In contrast, transcripts of some highly stable “housekeeping” genes, including ribosomal proteins, showed a de-enrichment of m^6^A [9]. However, in pathological situations, some m^6^A sites may be regulated in a disease-specific manner. In various cellular stresses, the investigators also observed changes in m^6^A levels in the 5’UTR as well [10]. Thus, the role of m^6^A modifications in different diseases and the role they play remains to be elucidated.

In recent years, the role of m^6^A modifications in cancer has received increasing attention as epigenomics and oncology studies continue to progress. It has been found that m^6^A modifications in tumors can regulate the stability, splicing, nuclear translocation, and translation efficiency of various mRNAs [11,12], which in turn leads to a complex series of molecular events. m^6^A is added to RNA by the m^6^A writer-complex which includes METTL3, METTL14, WTAP, VIRMA, RBM15/15B, ZC3H13, and CBLL1 [13,14]. m^6^A readers include YTHDF1/2/3, YTHDC1/2, IGF2BP1/2/3, HNRNPC/G/A2B1 [13]. They act as RNA binding proteins to exert their effect on the RNA life cycle subsequently. m^6^A erasers, ALKBH5 and FTO, are demethylases that can remove m^6^A from RNA. As the key catalytic subunit forming m^6^A modification, there is increasing evidence in recent years that the m^6^A writer METTL3 is significantly aberrantly expressed in tumors and can play a key role as an oncogene in most cases, leading to different phenotypic changes in tumors, resulting in proliferation, invasion, metastasis, and drug resistance. For example, in bladder cancer, METTL3 is significantly overexpressed and is associated with proliferation, invasion, and tumorigenic capacity in in vivo, and METTL3 promotes tumor progression through m^6^A modification on AFF4 and NF-κB mRNA, which in turn activates MYC transcription [15]. In hepatoblastoma, abnormally high expression of METTL3 leads to a significant increase in m^6^A levels in the tumor, and m^6^A is enriched not only near the mRNA stop codon but also in the coding sequence (CDS) region. The elevated stability of CTNNB1 due to its m^6^A modification leads to significant activation of the Wnt/β-catenin signaling pathway, which in turn promotes malignant proliferation of tumors [16]. In esophageal cancer, METTL3 is also significantly overexpressed and can lead to mRNA degradation by upregulating the m^6^A level of APC mRNA and recruiting the m^6^A “reader” protein YTHDF2. The reduced expression of APC leads to abnormal activation of the Wnt/β-catenin signaling pathway, thus promoting the glycolytic process and malignant cell proliferation in tumors [17]. Interestingly, sometimes METTL3 also acts as a tumor suppressor. Cui et al. [18] reported that knocking down METTL3 altered m^6^A enrichment on ADAM19 and promoted the malignancy of glioblastoma stem cells. Wu et al. revealed that METTL3 mediated m^6^A modification of FBXW7 and suppressed the development of lung adenocarcinoma subsequently [19]. Thus, METTL3 regulates the fate of these RNAs through m^6^A modification at key transcripts, which in turn affects the development of many cancers, including hematologic malignancies and solid tumors.

## 2. Aberrant Translation in Cancer

In the life cycle of mRNA, translation, the process of protein synthesis, is the most energy-consuming step in the entire cell [20], and this step plays a key role in the regulation of gene expression. With the rapid development of high-throughput sequencing technologies in recent years, mathematical modeling and multi-omics analysis have revealed that the magnitude of translational regulation in cells exceeds the sum of transcription, mRNA degradation, and protein degradation [21]. Components of the translational machinery integrate almost all oncogenic signals [22], and dysregulation of the translational process is considered one of the hallmarks of tumors and is associated with abnormal proliferation, angiogenesis, differentiation, and immune response [23,24]. Aberrant mRNA translation is a common feature of tumors, in which the process cannot be separated from the involvement of RNA-binding proteins (RBPs) in canonical translation machinery, including eukaryotic initiation factor (eIF) and elongation factor (eEF) (Figure 1). Their signals are aberrantly amplified in tumors [25,26].

### 2.1. Translation Factors in Eukaryotic Translation

Translation of mRNA in eukaryotic cells includes cap-dependent and cap-independent translation, in which the eIF4F complex plays an important role. eIF4F contains three components (eIF4E, eIF4G, eIF4A), of which eIF4E is the cap-binding subunit of the eIF4F complex and is required for cap-dependent translation of all nuclear-encoded mRNAs [27]. In addition, eIF4E can also stimulate the RNA unwinding enzyme activity of eIF4A independently of its cap-binding function and thus promotes translation [28]. eIF4E interacts with eIF4G and binds the m7G cap structure of mRNA, which in turn promotes translation. eIF3 plays a central role in the translation initiation of classical cap-dependent translation and cap-independent translation [29,30,31]. Different subunits of eIF3 confer different functions to the eIF3 core complex. Besides, other eukaryotic initiation factors also play important role in the translation process. Translation initiation is generally regulated by the 43S pre-initiation complex (43S PIC), which consists of eIF1, eIF1A, eIF3, eIF5, and the ternary complex (TC) [22]. The TC is formed by eIF2 (containing α, β, γ subunits), tRNA, and GTP. When eIF2α is phosphorylated under stress, the TC formation is inhibited and the global translation is downregulated subsequently [32,33]. eIF6 was first reported to participate in the biogenesis of the 60S ribosomal subunit in the nucleus as an anti-association factor [34,35]. However, Gandin et al. [36] found that eIF6 is rate-limiting for efficient translation initiation. In the cytoplasm of mammalian cells, the phosphorylation of eIF6 on Ser235 leads to its release from the 60S, which promotes the formation of a translation-competent 80S ribosome [37]. The translation elongation process is carried out by the ribosome with the assistance of eEFs. Among them, eEF1A is an important component of the translational apparatus as it interacts with tRNA [26]. eEF2 possesses an RNA binding site that interacts with tRNAs and promotes conformational changes, thus allowing the latter to interact with the coding region of mRNAs, mediating the translocation of peptide chains in extension to the P-site of the ribosome [38]. 

### 2.2. Dysregulation of Translation Factors in Cancer

Most of the eIF4E-sensitive mRNAs have a long and highly structured 5’UTR region [39], and these mRNAs encode many proteins associated with cell proliferation and tumor progression, including MYC, VEGF, cyclin, and others [40]. Many studies have reported that overexpression of eIF4E is associated with poor prognosis in cancer patients and can lead to tumor vascularization and invasion [41]. In recent years, various subunits of eIF3 have been found to have altered expression in malignant tumors, affecting translation of oncogenic mRNAs. eIF3a expression level were first found to be elevated in breast cancer tissues compared to paired normal breast tissues by Bachmann et al. [42], and eIF3a might play an important role in regulating translation of specific mRNAs encoding α-microtubulin, RRM2, and proteins associated with the cell cycle [43]. In virus-induced murine mammary tumors, the eIF3e gene was identified as a common insertion site and suggested that production of truncated eIF3e could lead to malignant transformation of mammary epithelial cells [44]. In prostate cancer, elevation of eIF3h positively correlates with tumor stages. The expression level of eIF3h is higher in metastatic prostate cancer than in primary prostate cancer, and eIF3h may play an important regulatory role in the translation of specific mRNAs [45]. Therefore, aberrant overexpression of eIF3h may contribute to tumor development by upregulating the translation of important mRNAs associated with cell proliferation [46]. Other translation factors are also dysregulated in cancer. eIF1 expression is downregulated in pancreatic ductal adenocarcinoma [47]. eIF1A is essential for cell proliferation and the cell cycle in cancer [48]. Interestingly, although eIF2α phosphorylation leads to reduced global translation, the translation of a restricted subset of mRNAs is enhanced, which facilitates glycolysis and cell invasion in cancer [49,50]. eIF5 is overexpressed in colorectal cancer and hepatocellular carcinoma and predicts poor prognosis [22,51]. eIF6 is reported to be markedly upregulated in hepatocellular carcinoma, colorectal cancer, and gallbladder cancer, which lead to tumor progression via mTOR and AKT-related signaling pathways [52,53,54,55]. Targeting eIF6-mediated translation blunts lipid accumulation and oncogenic transformation in the liver [56]. eEF1A plays an important and well-defined role in cancer development and progression [26,57], and eEF1A is aberrantly highly expressed in a variety of tumors and suggests a poor prognosis [58,59]. eEF2 also plays an important role in promoting the progression of tumors such as breast cancer [38,60,61]. Taken together, these translation factors affect the translation initiation and elongation process of multiple mRNAs in different types of cancer.

As a key enzyme regulating mRNA fate, the regulation of METTL3 on the translation process in tumors cannot be ignored. In recent years, many studies have reported that METTL3 could lead to changes in the expression of target genes through the regulation of mRNA translation process, which in turn caused tumor progression. The importance of RBPs in canonical translation machinery in the aberrant translation of m^6^A-modified mRNAs was also mentioned in many studies [7,62]. Therefore, the following is intended to introduce the various mechanisms involved in the translational regulation of mRNA by METTL3 in cancer and the interconnection between various m^6^A reading proteins and canonical translation machinery in these processes, so as to provide new ideas and possibilities targeting METTL3-mediated translation regulation for cancer treatment.

## 3. METTL3 Functions as a Translation Regulator in Cancer

The life cycle of m^6^A-modified mRNA begins with the transcriptional process, and m^6^A modifications are mainly mediated by METTL3 in the nucleus. In general, when mRNA is transported to the cytoplasm, specific m^6^A reader proteins bind to m^6^A, which in turn affects the translation of mRNA [63]. Many studies in recent years have found that METTL3 could be directly or indirectly involved in the translational regulation of mRNAs through a variety of mechanisms, as summarized in Table 1.

### 3.1. METTL3 Mediates the Binding of Other m^6^A Reader Proteins to Promote Translation through m^6^A Modification

Wang et al. reported in 2015 that after m^6^A modification of the 3’UTR and stop codon regions of mRNA by METTL3 in the nucleus, mRNA translocated to the cytoplasm, where the m^6^A reader YTHDF1 selectively recognized the m^6^A sites. Eukaryotic translation initiation factors were recruited by YTHDF1 subsequently, which in turn promoted ribosome loading and assembling on target mRNA, and advanced cap-dependent or cap-independent translation initiation [7,64].

Subsequently, a large number of studies have emerged to support and enrich this theory. Song et al. discovered that in colorectal cancer, METTL3 catalyzed m^6^A modification in 3’UTR of HSF1 mRNA and protein expression of HSF1 was significantly downregulated after knockdown of YTHDF1, demonstrating that METTL3-mediated m^6^A could promote translation through the m^6^A reader YTHDF1 [65]. In endometrial cancer, reduced METTL3 expression leads to a decrease in m^6^A modification on PHLPP2 mRNA, resulting in attenuated YTHDF1-mediated translation of PHLPP2, which in turn caused de-repression of the AKT pathway and promotes tumor progression [66]. In melanoma, knockdown of METTL3 in bone marrow cells results in the lack of m^6^A modification on SPRED2, which in turn disrupts YTHDF1-mediated mRNA translation, leading to enhanced activation of NF-κB and STAT3 via the ERK pathway, contributing to tumor progression [67]. In gastric cancer, METTL3-mediated SPHK2 m^6^A modification followed by YTHDF1 facilitates translation initiation through interacting with eIF3a, which in turn upregulates the translation efficiency and promotes tumor progression [68]. In lung adenocarcinoma, m^6^A modification promotes YTHDF1-mediated translation of ENO1 and SLC7A11, thereby enhancing tumor glycolysis and ferroptosis [69,70]. In ocular melanoma, down-regulated METTL3-mediated m^6^A modification leads to attenuated YTHDF1-mediated translation of tumor suppressor HINT2, which promotes tumor progression [71]. In hepatocellular carcinoma, researchers discovered that METTL3 could catalyze m^6^A modification in CDS and 3’UTR of SNAI1 mRNA, and YTHDF1 tended to bind to m^6^A in CDS of SNAI1 to mediate translation. Researchers then treated cells with rapamycin and found that YTHDF1 mediated the cap-independent translation of SNAI1 and that YTHDF1 could synergize with eEF2 to promote translation extension of SNAI1 mRNA [72]. In gastrointestinal stromal tumors, METTL3 recruits YTHDF1 through m^6^A modification of MRP1 mRNA in 5’UTR and promotes translation extension by eEF1, which in turn leads to intracellular translocation of MRP1 to imatinib and promotes drug resistance [73]. In cervical and liver cancer, METTL3 mediates m^6^A modification of PDK4 in 5’UTR, and subsequently YTHDF1 synergizes with eEF2 to promote the translation of PDK4, which in turn enhances the glycolytic process in tumors [74]. In breast cancer, METTL3 mediates the m^6^A modification of KRT7 in CDS, followed by enhanced translation elongation with the involvement of YTHDF1/eEF1 [75].

In addition to YTHDF1, METTL3 can also regulate translation of mRNAs through YTHDF3. Researchers identified a certain overlap between YTHDF3 and YTHDF1-bound proteins in the cytoplasm, and it was found that YTHDF3 and YTHDF1 could simultaneously interact with eIF4A, which in turn accelerated the translation process. It was suggested that the m^6^A reader YTHDF3 could enhance YTHDF1-mediated translation after METTL3-mediated m^6^A modification in some mRNAs [64]. This mechanism was validated in the subsequent studies: METTL3 could promote translation initiation complex formation through m^6^A modification of YAP mRNA in lung cancer by recruiting YTHDF1/3 as well as eIF3b, which in turn improved the translation efficiency and stability of YAP mRNA [76], resulting in tumor metastasis. In bladder cancer, after m^6^A modification of ITGA6, YTHDF1 cooperated with YTHDF3 to promote the translation of ITGA6 and mediated tumor progression [77].

METTL3 also enhances mRNA translation through other m^6^A readers. Liu et al. found that METTL3 mediated m^6^A modification in 3’UTR of BMI1 in oral cancer [78]. Overexpression of METTL3 increased the binding of BMI1 mRNA to polysomes without altering the stability of BMI1 mRNA and the rate of protein degradation, suggesting an enhancement of the translation process. Subsequently, to investigate through which m^6^A reader protein METTL3 promotes translation, authors knocked down IGF2BP1, IGF2BP2, IGF2BP3, and YTHDF1 and found that BMI1 mRNA expression was not altered, while knockdown of IGF2BP1 downregulated BMI1 protein level. The above experiments illustrated that METTL3 could also promote the translation of mRNAs such as BMI1 through m^6^A readers other than YTHDF1/3, such as IGF2BP1. In esophageal cancer, METTL3 mediates m^6^A modification in 3’UTR of TNFR1 mRNA [79]. ATXN2 acts as a novel RNA binding protein and enhances the translation of m^6^A-modified TNFR1 mRNA.

### 3.2. METTL3 Enters the Cytoplasm to Facilitate the Translation Process

In oncology studies, researchers found that after m^6^A modification in the 3’ UTR of a large subset of mRNAs at sites close to the stop codon, METTL3 itself could tether to the mRNA as an m^6^A reader in the cytoplasm. Subsequently, METTL3 formed as a “bridge” between the 3’ UTR and the 5’cap-binding proteins of mRNA, which supported an mRNA looping mechanism for ribosome recycling and translational control. In addition, the researchers observed the close proximity of METTL3 and individual polyribosomes with cap-binding proteins such as eIF4E by electron microscopy, and found that METTL3 and eIF3h had direct physical and functional interactions, thus promoting the translation of a large number of oncogenic mRNAs including BRD4 [62,80].

Several studies subsequently reported evidence for the direct involvement of METTL3 in translation regulation in the cytoplasm. Song et al. found that METTL3 deletion in colorectal cancer significantly reduced the level of HSF1 mRNA in the polyribosome fractions and increased its level in the non-translating ribosome fractions. m^6^A-modified HSF1 resulted in direct tethering of METTL3 to HSF1 mRNA in the cytoplasm to facilitate the translation process [65]. In addition, interestingly, miR455-3p could also inhibit the translation of HSF1 mRNA by interacting with the m^6^A site of HSF1 located in 3’UTR, preventing METTL3-mediated m^6^A modification as well as the direct binding. In cervical cancer, researchers found that METTL3 was mainly localized in the cytoplasm. Knockdown of YTHDF1 did not affect protein expression of AXL, but protein expression of AXL was significantly upregulated after overexpression of both wild-type or catalytic mutant METTL3, suggesting that METTL3 was directly involved in the translation process of AXL in the cytoplasm [81]. In chronic myeloid leukemia (CML), investigators found the presence of METTL3 in the cytoplasm, then they verified that METTL3 knockdown led to a reduction in global translation efficiency in CML cells and showed a critical role for METTL3 in maintaining ribosome levels and translational potential [82]. Subsequent knockdown of METTL3 resulted in a significant decrease in m^6^A levels of genes involved in ribosome biogenesis and translation such PES1. After overexpression of wild-type and mutant METTL3 in cells, it was found that the protein levels of PES1 were both significantly increased, while the mRNA levels were unchanged, and both wild-type and catalytic mutant METTL3 were found to bind to 3’UTR of PES1 mRNA in the cytoplasm. Thus, this study revealed that METTL3 could promote the production of PES1 protein by directly binding to m^6^A-modified PES1 in the cytoplasm, which in turn upregulated the translation efficiency of its mRNA, ultimately allowing for enhanced ribosome synthesis and translation processes of other oncogenic mRNAs. In bladder cancer, investigators found that m^6^A levels in the 3’UTR of CDCP1 mRNA were increased during malignant transformation [83,84]. Mechanistically, after METTL3 mediated the m^6^A modification of CDCP1, METTL3 cooperated with YTHDF1 to bind to the 3’UTR m^6^A site of CPCP1 and thus promoted translation. Overexpression of METTL3 had no effect on the expression level and stability of CPCP1 mRNA, but upregulated the protein level of CDCP1 without changing the protein degradation rate, and significantly upregulated the polysome-bound CPCP1 mRNA. The catalytic mutant METTL3 can also promote the translation of m^6^A-modified CDCP1 mRNA, although with weaker activity compared with the wild type METTL3.

In 2022, Wei et al. [85] revealed a novel m^6^A-independent mechanism for METTL3 to regulate translation in gastric cancer progression. Cytoplasm-anchored METTL3 can promote the looping of some non-m^6^A-modified oncogenic mRNAs by interacting with PABPC1 and eIF4F complex. This study assigned a new function to cytoplasmic distributed METTL3 and expanded the ways in which METTL3 facilitates translation. We look forward to further studies to provide more evidence for this important and interesting finding.

### 3.3. Promoter-Bound METTL3 Enhances Translation

In addition to METTL3’s ability to upregulate translation efficiency through the binding of other reading proteins or binding to m^6^A-modified mRNAs directly, researchers have identified a mechanism by which METTL3 promotes m^6^A-dependent translation regulation through binding to the promoter of target genes. They found that in acute myeloid leukemia (AML), METTL3 could be localized to the transcription initiation site of target genes in chromatin independently of METTL14 which was essential for m^6^A modification [86]. Since the majority of target genes had the CAATT-box binding protein CEBPZ at the transcription initiation site, METTL3 could induce m^6^A modifications within the CDS region of related transcripts such as SP1 and SP2 mRNAs after their transcription by interacting with the CEBPZ protein and binding to these transcription initiation sites on chromosomes. Researchers found that the transcripts of METTL3-bound target genes were enriched in [GAG]n sequences that could cause ribosomal arrest during translation. When these sequences were modified by METTL3-mediated m^6^A methylation, ribosomal stalling was lifted, thereby contributing to enhanced translation. As transcription factors, SP1 and SP2 proteins played an important role in promoting AML progression. The transcriptional activity of SP1 and SP2 was unaffected by METTL3 deletion, but due to the lack of m^6^A modification on the transcripts, the transcripts were shifted to low molecular weight polyribosomes, resulting in reduced translational efficiency and less protein production, ultimately leading to reduced malignancy of AML cells.

### 3.4. Other Possible Pathways

Protein translation usually begins with the recruitment of the 43S pre-initiation complex to the 5’ cap structure of the mRNA via the cap-binding complex. However, some transcripts can be translated in a cap-independent manner through certain mechanisms. A study found that a single 5’UTR m^6^A could directly bind eIF3, which in turn allowed it to recruit the 43S complex to initiate translation without the involvement of the cap-binding protein eIF4E. The inhibition of adenosine methylation also selectively reduced the translation efficiency of the 5’UTR m^6^A-modified mRNA. This study revealed that cells under different stresses induced a redistribution of m^6^A modifications at the transcriptome level and could generate a translation pattern resulting from a 5’UTR m^6^A modification that bypassed the involvement of the eIF4F complex with eIF3 as a novel m^6^A reading protein [87]. Meanwhile, a similar mechanism was reported: in mammalian cells, the asymmetric distribution of m^6^A along mRNA resulted in relatively little methylation in the 5’UTR. However, in the heat shock stress response, certain adenosines in the 5’UTR of the newly transcribed mRNA were preferentially methylated, and increased m^6^A modification in this region promoted cap-independent translation initiation, revealing a novel mechanism of translation regulation under stress [88]. Therefore, it is worthwhile to further explore whether this mechanism of mRNA translation with eIF3 directly as an m^6^A reader involved in 5’UTR m^6^A modification exists in cancer.

In summary, METTL3 is involved in translation regulation in a variety of ways in cancer, which can be summarized as follows (see Figure 2):(1)METTL3 modifies m^6^A in target mRNAs and then recruits canonical translation machinery through classical or novel m^6^A readers YTHDF1/YTHDF3/IGF2BP1/ATXN2 to promote translation.(2)After m^6^A modification of target mRNAs, METTL3 directly binds m^6^A sites in the cytoplasm and recruits canonical translation machinery to promote translation.(3)METTL3 interacts with CEBPZ to bind the promoter of target genes and enhances the translation of the associated mRNAs by relieving ribosomal arrest through m^6^A modification.

## 4. Conclusions and Perspectives

In recent years, more and more studies have reported that METTL3-mediated m^6^A modification of mRNAs affected various processes in the mRNA life cycle through multiple intermolecular interactions, which in turn mediated various phenotypic changes in cancer. In this review, we focused on the involvement of METTL3 in the regulation of mRNA translation. First, we briefly introduced the aberrant translation in cancer and its relationship with canonical translation machinery. Then, we introduced in detail the different roles of METTL3 in the aberrant translation process of m^6^A-modified mRNA. With the gradual clarification of the mechanism of aberrant translation regulation in cancer, targeting the METTL3-mediated translation process has become a possibility. Since METTL3 interacts with multiple translation-related RBPs in the process of translation promotion, the use of small molecule inhibitors targeting canonical translation machinery for cancer treatment seems possible. Small molecule inhibitors targeting eIF4A and eIF4E have been introduced for a long time. For example, 4EGI-1, an inhibitor of eIF4E- eIF4G interaction [89], can decrease eIF4E-sensitive mRNA translation level and demonstrate a favorable antitumor effect. However, advancing to clinical application has been delayed [89,90]. The inhibitors against the binding of eIF6 to the 60S were also identified and might have dose- and cell-specific effects [91]. We cannot help but wonder whether targeting key molecules upstream of the translation machinery in cancer translation regulation could produce a broader spectrum and better efficacy of inhibitory effects on aberrant translation.

Encouragingly, the first launch of STM2457, a small molecule inhibitor targeting METTL3, was published in 2021, and its promising antitumor effect was validated in AML [92]. Moreover, Du et al. [93] used virtual screening of 1042 natural products and identified quercetin as a qualified METTL3 small molecule inhibitor. Moroz et al. [94] developed a METTL3 inhibitor UZH1a using a structure-based drug discovery approach. Lee et al. [95] reported eltrombopag as an allosteric inhibitor of the METTL3-14 complex. Since the determinants of successful clinical application of small molecule inhibitors include confirmation of the compound-mechanism hypothesis, compound-target action and pharmacodynamic activity [96], if the likelihood of successful drug development is to be maximized, a high standard of screening for inhibitors targeting translational regulation is required in preclinical studies. Therefore, whether drugs such as STM2457 can target METTL3 and thereby inhibit aberrant translation in other types of cancer such as lung cancer remains to be explored and validated through numerous studies. Combining small molecule inhibitors of METTL3 and canonical translation machinery inhibitors to overcome intra-tumor heterogeneity and inhibit the oncogenic signal of aberrant translation integration in cancer cells is also promising. Since the initial success of therapeutic approaches targeting abnormal translation in cancer has been achieved, we believe that in the near future, these drugs will move from the laboratory to the clinic and achieve breakthroughs in anti-cancer therapy.

## Figures and Tables

**Figure 1 biomolecules-13-00243-f001:**
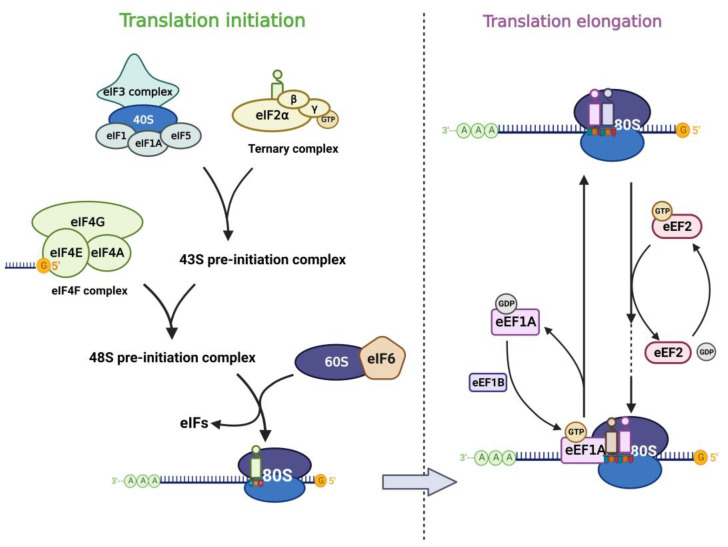
Translation factors participate in the mRNA translation process. The left panel shows the translation initiation process: eIF2 subunits combine with initiator methionyl tRNA and GTP to form the ternary complex (TC). TC associates with the 40S ribosomal subunit complex which consists of eIF3, eIF1, eIF1A, and eIF5 to form the 43S pre-initiation complex (43S PIC). Then 43S PIC is recruited to the mRNA template by combining to eIF4F complex and they form the 48S pre-initiation complex (48S PIC). The phosphorylation of eIF6 allows the 60S ribosomal subunit to join the 40S subunit, which leads to the formation of the translation-competent 80S ribosome and marks the end of translation initiation. The right panel shows the translation elongation cycle: eEF1A-GTP helps to deliver the aa-tRNA to the Aminoacyl site in ribosome and eEF1B recycles the released eEF1A-GDP subsequently. eEF2-GTP mediates the translocation of the elongating peptide to the Peptidyl site of the ribosome.

**Figure 2 biomolecules-13-00243-f002:**
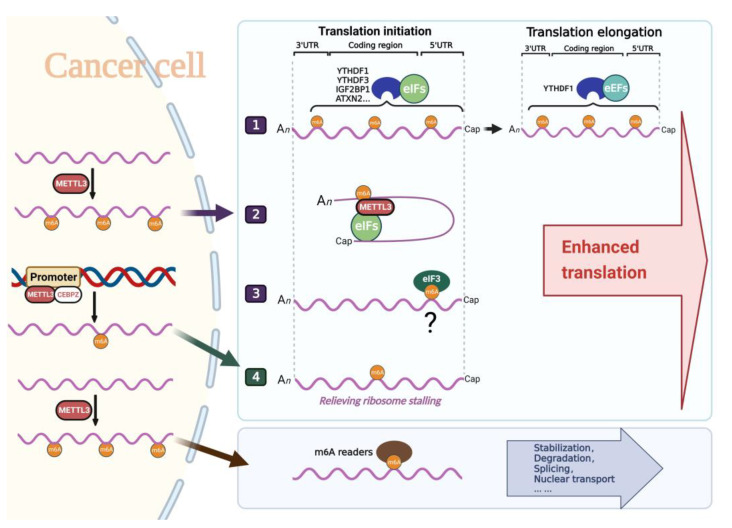
Molecular mechanisms underlying translation regulation of METTL3 on mRNAs in cancers. METTL3 methylates target mRNA transcripts in the nucleus and enhances its translation in the following ways: 1. METTL3 methylates target mRNA transcripts and recruits canonical translation machinery through classical reader proteins YTHDF1/YTHDF3/IGF2BP1 or novel reader protein ATXN2 to enhance translation. 2. METTL3 directly binds to 3’UTR of methylated mRNA in the cytoplasm and recruits canonical translation machinery to promote translation. 3. METTL3 methylates target mRNA transcripts and eIF3 binds to 5’UTR m^6^A to enhance translation in a cap-independent manner, but this mechanism has not been reported in cancer cells. 4. METTL3 interacts with CEBPZ to bind the promoter of target genes and relieves ribosome stalling through m^6^A modification to enhance translation.

**Table 1 biomolecules-13-00243-t001:** Mechanisms of METTL3 involved in translation regulation through m^6^A modification in human cancers.

PMID	Cancer Types	Reader Proteins *	Targets	Function	m^6^A Sites	RBPs of Canonical Translational Machinery
31061416	Liver cancer	YTHDF1	SNAI1	Epithelial-mesenchymal transition	CDS	eEF2
35032557	Gastrointestinal stromal tumor	YTHDF1	MRP1	Drug resistance	5′UTR	eEF1
32444598	Cervical and liver cancer	YTHDF1	PDK4	Glycolysis	5′UTR	eEF2
33795252	Breast cancer	YTHDF1	KRT7	Metastasis	CDS	eEF1
33654093	Melanoma	YTHDF1	SPRED2	Tumor growth and metastasis	CDS, 3′UTR	
30154548	Endometrial cancer	YTHDF1	PHLPP2	Proliferation and tumorigenicity		
33758320	Gastric cancer	YTHDF1	SPHK2	Progression		eIF3a
35078505	Lung adenocarcinoma	YTHDF1	ENO1	Glycolysis and tumorigenesis		eIF3e
31722709	Ocular melanoma	YTHDF1	HINT2	Progression	3’UTR	
34996469	Lung adenocarcinoma	YTHDF1	SLC7A11	Progression		
33618740	NSCLC	YTHDF1, YTHDF3	YAP	Drug resistance and metastasis	3’UTR	eIF3b
31409574	Bladder cancer	YTHDF1, YTHDF3	ITGA6	Progression	3’UTR	
32621798	Oral squamous cell carcinoma	YTHDF1, IGF2BP1	BMI1	Tumorigenesis and metastasis	3’UTR	
32838807	Colorectal cancer	METTL3, YTHDF1	HSF1	Progression	3’UTR	
30796352	Bladder cancer	METTL3, YTHDF1	CDCP1	Chemical carcinogenesis	3’UTR	
27117702	Lung cancer	METTL3	EGFR, TAZ	Progression	3’UTR	eIF4E, eIF3
30232453	Lung cancer	METTL3	BRD4	Tumorigenesis	3’UTR	eIF3h, eIF4E
34561421	Chronic myeloid leukemia	METTL3	PES1	Proliferation and drug resistance	3’UTR	
30249526	Ovarian carcinoma	METTL3	AXL	Epithelial-mesenchymal transition		
34995801	Esophageal cancer	ATXN2	TNFR1	Progression and tumorigenesis	3’UTR	
29186125	Acute myeloid leukemia		SP1,SP2	Proliferation	CDS	
34631715	Kidney cancer		ABCD1	Progression	5’UTR	
33676554	Lung adenocarcinoma		FBXW7	Apoptosis and proliferation	CDS	
34530048	Melanoma		EGFR	Drug resistance	3’UTR	
33217448	Colorectal cancer		GLUT1	Progression	3’UTR	
28920958	Acute myeloid leukemia		c-MYC, BCL-2, PTEN	Proliferation	3’UTR	
33267838	Bladder cancer		CDCP1	Progression	3’UTR	
31454538	Breast cancer		BCL-2	Progression		

* The reader proteins here refer to the classical or novel m^6^A readers as well as METTL3, which can also act as an m^6^A reader directly in some cases.

## Data Availability

The data that support the findings of this study are available from the corresponding author upon reasonable request.

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
