# Peer review of "Critical Roles of METTL3 in Translation Regulation of Cancer"

_biomolecules, 2023, doi:10.3390/biom13020243_

Round 1

Reviewer 1 Report

Here Meng and colleagues review in a concise and easy-readable manner part of the literature regarding METTL3, underlyning it's role as an important regulator of translation during tumorigenesis.

Despite being well written and sufficiently easy to follow, I would reccomend a few modifications/implementations, expecially regarding the literature which is cited and on how the review is overall presented 

More in detail:

-Section 1, Introduction) 

A comment on the fact that METTL3 can act as a tumor suppressor in some cases should also be added and briefly discussed (e.g. https://doi.org/10.1186/s13045-020-00951-w). A few words on the methyltransferase complex should be spent (METTL3 interactor METTL14 is not even mentioned)

METTL3 affects mRNA fate in several ways. As the authors say, it "can regulate the stability, splicing, nuclear translocation, and translation  efficiency of various mRNAs". The authors cite a few works in which METTL3 regulates stability of mRNAs, hence focusing on translation-independent roles of METTL3. I suggest adding a cartoon or table summarizing the general roles of METTL3 in order to make these points more clear to the reader.

-Section 2, Aberrant translation in cancer) 

Before getting in the business of describing METTL3 role, the authors talk about aberrant translation in cancer. They talk about the eIF4F complex, they enter in the details of eIF3, but they do not consider at all another really important IF: eIF6. I would suggest adding a brief section also on eIF6, which is widely known to be involved in cancer onset, citing at least a few recent works (e.g.  doi: 10.1186/s12967-021-02877-4., doi: 10.3390/ijms23147720., 10.1038/s41467-021-25195-1, doi: 10.1166/jbn.2019.2792., doi: 10.1007/s00432-019-03030-x). Accordingly I would add eIF6 in figure 1.

-Section 3, METTL3 functions as a translation regulator in cancer) 

Here the authors get in the topic, citing selected articles describing METTL3 role in regulating translation in cancer. The authors summarize METTL3 role in traslation in three major points and refer to figure 2. Such points are however not explicitated in the figure and thus conclusions are hard to follow. Also the legend for figure 2 does not seem to be explicative, I suggest re-writing it after addition of the 3 points in the figure

-4, Conclusions and perspectives)

Here authors draw their conclusions and propose how METTL3 could be targeted for cancer therapy.  The authors however did not cite some works which are important for making their point on targettability of METTL3 (examples include: https://doi.org/10.3389/fphar.2022.878135,  https://doi.org/10.1002/cmdc.202100291, https://doi.org/10.3390/ph15040440) which i suggest to add.

Before talking about METTL3 targetability, to make their point, the authors describe the existence of inhibitors of eIF4A and eIF4E.  Once again, eIF6 inhibitors should also be added as as they were recently published (doi: 10.3390/cells9010172.)

Generally speaking i think that the section in which targetability of METTL3 is described should be inserted, with the above-mentioned implementations, in a stand-alone section prior to the conclusion section.

Author Response

    Dear reviewer, we gratefully thank you for your time spend making constructive remarks and useful suggestions, which has significantly raised the quality of the manuscript and has enable us to improve the manuscript. Each suggested revision and comment brought forward by you was accurately incorporated and considered. We provided point-by-point response and revisions to your comments following the required format.

Reviewer 2 Report

Meng et al provide a concise and succinct review on the critical roles of METTL3 on translational regulation in cancer. Authors first provide an introduction on m6A RNA modification followed by the examples of deregulations in the translation initiation and elongation factors in cancer. They then extend their review to METTL3-mediated translational regulation in cancer where they discuss both m6A-dependent and independent regulation. I believe that the presented review could be highly useful to the scientists working in the field of epitransciptomics. I suggest the following revisions before considering the manuscript for acceptance:

1. It would be nice to have a separate subheading covering the m6A machinery in Introduction. Please introduce "writers, readers, and erasers" briefly followed by a more detailed information on writers.

2. I would suggest dividing "2. Aberrant translation in cancer" into two subheadings in which to introduce eukaryotic translation followed by deregulation in cancer.

3. Figure 1: Eukaryotic initiation factors interacting with a ribosome that is almost towards the end of a mRNA is misleading. eIF2/3/4 ..etc are assembled at the 5'-cap and stays with the small subunit of a ribosome until the translation initiation codon is recognized. Please revise the figure to prevent potential confusion. Additionally, the same figure representing both translation initiation and elongation add further confusion. Please prepare different mRNP complexes to depict the different stages of translation in this figure.  

4. "Aberrant translation in cancer" is missing eIF2-alpha and its phosphorylation. It would be nice to cover this topic as well.

Author Response

(The authors gave the same response as above.)
